# CrowdSFL: A Secure Crowd Computing Framework Based on Blockchain and Federated Learning

**Ziyuan Li**, **Jian Liu ***, **Jialu Hao ***, **Huimei Wang and Ming Xian**

College of Electronic Science and Technology, National University of Defense Technology, Changsha 410073, China; lzy3220001@gmail.com (Z.L.); freshcdwhm@163.com (H.W.); qwertmingx@sina.com (M.X.)

*   Correspondence: ljabc730@gmail.com (J.L.); haojialu.nudt@gmail.com (J.H.); Tel.: +86-155-7483-1545 (J.H.)

**Abstract:** Over the years, the flourish of crowd computing has enabled enterprises to accomplish computing tasks through crowdsourcing in a large-scale and high-quality manner, and therefore how to efficiently and securely implement crowd computing becomes a hotspot. Some recent work innovatively adopted a P2P (peer-to-peer) network as the communication environment of crowdsourcing. Based on its decentralized control, issues like single-point-of-failure or DDoS attack can be overcome to some extent, but the huge computing capacity and storage costs required by this scheme is always unbearable. Federated learning is a distributed machine learning that supports local storage of data, and clients implement training through interactive gradient values. In our work, we combine blockchain with federated learning and propose a crowdsourcing framework named CrowdSFL, that users can implement crowdsourcing with less overhead and higher security. In addition, to protect the privacy of participants, we design a new re-encryption algorithm based on Elgamal to ensure that interactive values and other information will not be exposed to other participants outside the workflow. Finally, we have proved through experiments that our framework is superior to some similar work in accuracy, efficiency, and overhead.

**Keywords:** crowd computing; blockchain; federated learning; re-encryption algorithm

## 1. Introduction

The concept of crowdsourcing has come a long way in the 14 years since it was first proposed by Jeff Howe in 2006. It has been used to solve a number of real-world problems [1]. The earliest definition of crowdsourcing is the practice of a company or organization to outsource the tasks, usually performed by employees, to non-specific (and usually large) Internet crowds in a free and voluntary manner. Crowdsourcing tasks usually rely on human knowledge and intelligence. However, if it involves tasks that require many complex calculations, that is, crowd computing, it may additionally rely on computing devices like CPUs. At present, crowdsourcing is actively developing, ranging from the analysis of big data [2] to the location service [3]. At the same time, some enterprises such as CrowdSpring, Uber, and Upwork are working hard to explore new working models and layouts for crowdsourcing.

The rapid development of crowdsourcing has benefited from the success of his predecessor. In 1990, some researchers proposed that distributed computer be collected to complete a more complex task. Volunteer computing was proposed in that era and has achieved particular development. Many projects that rely on volunteer computing came into being. The first large-scale projects to do this and gain public appeal were prime number hunting by the Great Internet Mersenne Prime Search (GIMPS). Afterward, there appeared a distributed computing project Folding@home that aims at computing molecular dynamics trajectories of biological systems [4], and SETI@home that aims at

searching for signs of intelligent extraterrestrial life in radio signals. These projects are all realized with the help of idle computing resources on the Internet. However, due to the backwardness of computing power at that time, few of them have achieved considerable achievements. After that, the system framework of volunteer computing has been further improved [5], and one significant product under it is crowdsoucing.

The prosperity of crowdsourcing is a remarkable revolution, which has brought together people distributed all-over the Earth to complete knowledge-intensive tasks [2,3]. We are also currently in another crowd revolution, the crowd of computers. Early crowdsourcing mainly relied on manpower to solve tasks. With the increase in the computing power of PCs, laptops, and mobile phones, the strides of crowd computing are in an exploding developmental period [6]. With crowd computing, idle computing power is aggregated together to effectively solve some computation-intensive tasks.

Murray et al. [6] proposed crowd computing for the first time in their work and briefly introduced some cases. They proposed the idea of spreading the computational jobs to a collection of smartphones using an opportunistic network. Today, crowd computing has different connotations depending on the perspectives. However, as a type of crowdsourcing, four basic components will always be required:

- **Requester.** Typically, it is a server that hosts the crowd task. It is generally responsible for standardizing a task before releasing it. In some special tasks, it can even participate in the project as a unique worker.
- **Workers.** Crowdworkers lend their devices to execute the jobs. In crowd computing, the entities of workers are generally idle devices contributed by crowdworkers, such as mobile phones or laptops. When a worker chooses to participate in a crowdsourcing task, their devices will execute the tasks according to an agreement. When the devices are found to be idle, the jobs are processed following the (CPU) cycle-stealing scheme.
- **Crowdsourcing Platform.** Middleware for job management along with the server and client applications. The server application is responsible for creating tasks, discovering suitable workers, assigning tasks and scheduling tasks to the designated workers. Collecting the results from multiple workers, assembling them, and updating it in the server application, and for additional purposes. The platform will handle a large amount of work, so it usually needs to have strong computing performance or be deployed on the cloud.
- **Network.** All devices communicate through the Internet or a local area network. Traditional crowdsourcing applications generally uses the Internet as the network, and a platform as the center to control all devices. This paper innovatively proposes a blockchain-based crowdsourcing paradigm, and the essential explanation is described in Section 4.

Centralized crowdsourcing layouts often expose many security issues. In October 2016, Uber was faced with a serious privacy leak scandal. Hackers utilized the bug in the permission access program to steal the privacy data of 57 million users on the server. Uber could only temporarily disable access to the data and paid a huge ransom to the hackers [7]. In May 2014, the famous crowdsourcing application Upwork suffered a severe DDos attack, which caused network abnormalities in several of its services, and its technical support services were temporarily interrupted [8].

The development of crowdsourcing systems is often limited by the security weaknesses of the centralized nature of the systems, and there have been many research efforts to solve them. Encryption and differential privacy [9–12] are used to protect the data privacy of all participants who involved in the communication process. Reputation-based mechanisms are proposed to address "false-reporting" and "free-riding" behavior [13,14]. Some researchers have proposed the idea of swapping the distributed network structure, like peer-to-peer network, to solve security issues and bottlenecks in a traditional crowdsourcing system [15]. Therefore, this research is motivated by a problem: Can a new structure be used to solve the security and privacy issues encountered in traditional crowd computing while ensuring that crowdsourcing remains efficient enough? To answer this question, we propose a blockchain-based decentralized framework that uses both re-encryption

algorithm and federated learning methods. The framework can improve user security and service availability, and greatly improve auditability. In previous crowdsourcing systems, a reward was given to the worker who proposed the best solution [5]. In our design framework, the hardworking workers can get corresponding rewards based on their effort. Our contributions in this paper can be summarized as follows.

1. We transplant the entire crowdsourcing system onto a the blockchain, and each participant is an account in the blockchain. In order to make full use of the advantages of the blockchain, a data interaction mode controlled by smart contracts running on Ethereum is proposed. The code set in the contract enables the data to be uploaded and saved in blocks with the correct format.
2. We propose crowd computing with federated learning as the computing paradigm. Unlike previous paradigms, the interactive content of this crowdsourcing system is a value of the gradient, or model. Using the intermediate training value of federated learning as the interaction can effectively protect user privacy. At the same time, in each round of the federated learning, the gradient submitted by workers would be judged by the requester, and feedback to the platform for the distribution of rewards. Every worker who effectively participates in crowdsourcing can get a certain reward. This provides another innovative way for the reward distribution to future crowdsourcing systems. As far as we know, this is the first time that federated learning has been adopted into crowd computing, and our research is an innovation in crowdsourcing.
3. We propose a new re-encryption algorithm based on the blindness nature of Elgamal. With this algorithm, as long as requesters and the platform do not collude, the value of the gradient or the model submitted by the workers would not be leaked to the platform or requester. It enables the decentralized blockchain platform to ensure data confidentiality, anonymity, and accountability during data interaction. Our algorithm does not take advantage of ciphertext-homomorphism to protect privacy, so compared to some recent similar works on homomorphic encryption methods [16,17], it does not add too much pressure to the computation overhead.

The rest of the paper is organized as follows. The related work is discussed in Section 2. We introduce the main components in Section 3, including crowdsourcing, blockchain, federated learning, and the re-encryption scheme. According to our innovations, we formalize the design ideas of the framework in Section 4. The system model and the threat model of CrowdSFL that we designed are introduced in detail in Section 5. We analyzed the efficacy and overhead costs of CrowdSFL in processing real-world datasets, and compared it with similar work in Section 6. Finally we conclude and discuss the future work emanating from the present study in Section 7.

## 2. Related Work

### 2.1. Crowdsouring in Adversarial Settings

Crowdsourcing is a business model that liberates labor, most of which are used to solve low labor and high-cost tasks. For example, in the field of machine learning, training datasets are just as important as algorithms. A good model needs sufficient sample data for training. However, in the real world, most of the sample data are heterogeneous due to multiple data sources, and some data have different structures, which brings challenges to the development of training. Fortunately, this problem can be solved by crowdsourcing. Albarqouni et al. [2] introduced a new concept for learning from crowds that handle data aggregation directly as part of the learning process of the convolutional neural network (CNN) via an additional crowdsourcing layer (AggNet). When heterogeneous training data pass through AggNet, they will be transformed into data with similar attributes and structure. They also confirm that aggregation and deep learning from crowd annotations using the proposed AggNet is robust to "noisy" labels and positively influences the performance of their CNN in the refining phase.

Security issues are often exposed to crowdsourcing in adversarial settings, collusion, or some other malicious behaviors would have a terrible impact on crowdsourcing results and affect the distribution of benefits.

Team crowdsourcing in social networks provides a promising solution for complex task crowdsourcing, where the requester hires a team of professional workers that are also socially connected can work together collaboratively. In this scenario, selfish workers can manipulate the crowdsourcing market by behaving untruthfully. This dishonest behavior discourages other workers from participating and is unprofitable for the requester. Wang et al. [18] proposed threshold payment rules based on the workers' skills required for the task, compared to other state-of-the-art polynomial heuristics, the proposed large-scale-oriented mechanism can achieve truthfulness while generating better social welfare outcomes.

Spatial crowdsourcing may require the exact geographic location of each worker before the assignment of tasks. Yuan et al. [3] propose a privacy-preserving framework without online trusted third parties. They devise a grid-based location protection method, which can protect the locations of workers and tasks while keeping the distance-aware information on the protected locations.

### 2.2. Privacy Preserving in Distributed Computing

In recent years, lots of works have emerged on the dealing with data security in distributed computing, such as access control technology based on attribute-based encryption [9,19] and privacy preserving data mining methods [10,11,20]. Rong et al. [10] propose a collaborative kNN protocol base on a set of secure building blocks using the ElGamal encryption in multiple cloud environments. The protocol could hide data access patterns and is designed without bit-decomposition. Wu et al. [11] improved on this basis of [10], and utilized a re-encryption algorithm to solve the security of data access patterns and query result patterns in a dual-cloud environment, and experimentally proved that it achieved less communication consumption than [10].

The advent of federated learning (FL) has changed the privacy risks of traditional distributed deep learning systems that need to aggregate users' data. Interaction through gradients can reduce the possibility of malicious third parties obtaining user privacy. However, some works have proved that a malicious server can reverse user data through the vary of gradient in each round [21]. In order to solve this security issue, the training gradients submission and the gradients aggregation operations must be in an anonymous or encrypted way, like [22]. Ding et al. [23] designed a new fully homomorphic encryption system based on the Paillier encryption system and supports proxy re-encryption. This encryption system is specifically designed for multi-user ciphertext calculation in distributed systems, so Tang et al. [12] uses this encryption system to improve the work of [22], ensuring that if a malicious user colludes with the server, the information of other users will not be leaked while ensuring high training accuracy. However, the disadvantage of [12] is that the encryption system in [23] requires a large computational overhead, which results in that the communication consumption of [12] is almost double that of [22] when the same task is completed.

### 2.3. Blockchain Meets Traditional Schemes

In the past two years, many works have been done on the blockchain. The public realizes that blockchain as a distributed system has essential advantages in nature. A benefit from the consensus mechanism is its decentralized nature guarantees the integrity and security of the data.

One of the most exciting creations in the blockchain industry is Dapp. Dapp is essentially a smart contract deployed on a blockchain platform such as Ethereum. It cleverly combines consensus mechanism, hlso that the logical operations that could have been performed locally have been performed by the entire blockchain P2P network. Lu et al. [24] first attempted of decentralizing crowdsourcing system atop the blockchain in a privacy-preserving way. The framework named ZebraLancer in their paper supports anonymously releasing tasks and sending anonymous answers and reconstructing some protocols such as registration-authority, answer-collection, and reward in the

form of smart contracts. Li et al. [15] proposed a practical crowdsourcing platform based entirely on smart contracts, named CrowdBC. All the logical processes in the crowdsourcing, such as REQ releases tasks, workers submit solutions, CSP evaluates, distribution of rewards, etc., are all reconstructed in the form of smart contracts and deployed in the application layer of CrowdBC's three-layer-architecture. It [15] makes almost all the affairs of crowdsourcing public and transparent, and the data of the entire transaction is non-tamperable and traceable. While their work does not include the design of evaluation mechanisms, they explain that this is due to the realistic diversity of tasks, and the related contract design would change with the actual background.

The data saved by the blockchain can be guaranteed not to be tampered with. Zheng et al. [25] designed a scheme to protect the data integrity of the distributed cloud storage system. Each uploaded file is divided into multiple ciphertext blocks by the threshold Paillier algorithm. When uploading the ciphertext block to the cloud server, the cloud stores the hash of the ciphertext block in the blockchain. At the same time, the hash value in the blockchain would be compared to ensure that each sub-block has not been tampered with.

Blockchain can also be adopted in privacy preserving data mining. Kuo et al. [26] proposed a role-alternating framework called GloreChain. They transplanted the logistic regression task under the C/S architecture to the decentralized blockchain system and each client took turns to act sever for aggregating logistic regression weight. This mechanism can effectively overcome the single-point-of-failure, but his work has certain limitations. They did not consider the low-quality model that may be caused by nodes with large differences in data distribution or very high-dimensional datasets. In [16,17], Shen et al. applied the blockchain as a database platform for data aggregation. However, in terms of the application value of the blockchain, it does not support a large amount of data storage. The cost of storing huge data on the blockchain in [16,17] is hard to pay.

Preuveneers et al. [27] chose to upload the gradient instead of the entire dataset in the FL to the blockchain. Compared with huge datasets, the gradient data size is more compact, and it is far better than [16,17] in terms of practicality. The flaw in [27] is that Preuveneers do not consider the security issues of uploading gradient plaintext. Their work focuses on practical applications but does not take into account the privacy issues that may be caused.

Weng et al. [28] proposed the DeepChain, which gives mistrustful parties incentives to participate in privacy-preserving learning, share gradients, and update parameters correctly, and eventually accomplish iterative deep-learning with a win-win result. Their work is better than [27] in confidentiality, and [28] takes into account the security issues of gradient interaction. However, the threshold Paillier algorithm used in their work has high computational overhead, and the experiment does not fully explain the change in throughput caused by this algorithm. At the same time, the incentive mechanism based on Deepcoin they proposed was not shown in the experiment. Similar work also includes the LearningChain [29] proposed by Chen et al. The difference with Deepchain is that Learnchain adopts a differential privacy mechanism to protect data holders' data privacy in the local training process, and design the l-nearest scheme to defend Byzantine attacks in the global gradients aggregation process.

Inspired by previous related work, the CrowdSFL we proposed in this paper adopts the same storage strategy as [27], while using the original re-encryption algorithm to protect the confidentiality and anonymity of data.

## 3. Main Components

In this section, We introduce the main components used in the framework.

### 3.1. Crowdsourcing System

Generally, one crowdsourcing system consists of three participants: requesters (REQ), workers, and the crowdsourcing platform (CSP). Some users on the network may, due to computing resources, time constraints, data capacity, or other reasons, have challenges completing their task. These users

would rely on crowdsourcing to become REQ and release their tasks to the crowdsourcing platform. In a complete workflow, workers could submit their own solutions to the CSP, and then the REQ could choose the best solution for their task. Finally, CSP distributes the reward to the corresponding workers. For some significant tasks, REQ or workers need to submit a deposit to the CSP in advance to ensure that they will actively participate in a certain task.

At present, common crowdsourcing systems are centralized, which means crowdsourcing platforms set up certain protocols to achieve worker selection, task assignment, or distribution of the rewards. These platforms require REQ and workers to submit superabundant detailed information which would cause leakage of privacy. The integrity of crowdsourcing systems can easily be rewritten by hackers. In mature crowdsourcing systems, reputation mechanisms are generally used to provide a reference for requesters to select workers. Although crowdsourcing platforms will adopt the reputation mechanism or special incentive mechanism to ensure the fairness, the centralized system means that once the hacker attacks successfully, there will be disastrous consequences, in other words, severe security risks exist. Figure 1 illustrates a traditional centralized crowdsourcing system.

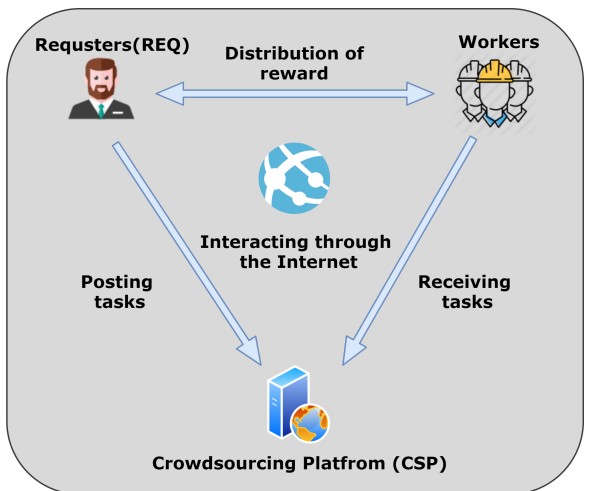

**Figure 1.** A traditional centralized crowdsourcing system.

Considering the possible threats to a centralized crowdsourcing system, distributed crowdsourcing systems are becoming the focus of current research. Zhang et al. [30] presented one distributed crowdsourcing system to achieve completing a computation task in a collaborative way and get the minimal makespan. To implement a federated learning paradigm for data security interaction, we also adopt a distributed crowdsourcing system, that differs from the distributed crowdsourcing system proposed in [30,31], because the data and information interaction in our system is based on blockchain-based architecture.

### 3.2. Federated Learning

Federated learning (FL) aims to solve this problem: it wants to keep the proprietary data of each user local, while the post-federated system can establish a virtual shared model through the exchange of parameters without violating data privacy regulations [32]. Its theoretical basis is based on machine learning, such as the loss function and multiple iterations. The difference is that it is a distributed machine learning, which requires a server acting as an aggregator to control all nodes. When the aggregation model is established, the data itself will not move, nor will privacy and data compliance be compromised.

In our framework, the clients of FL are the workers in a crowdsourcing system. They generally have spare capacity and idle computing power, in order to obtain financial or informational rewards, they go to compete tasks published by REQ. In addition, the clients' data would like to be non-IID and unbalanced that poses challenges to model aggregation and reward distribution in our crowdsourcing

systems. "FederatedAveraging" recently proposed in [33], is a proper algorithm to tackle these challenges. We also adopt this algorithm. Related pseudo-code is given in Algorithm 1.

---

**Algorithm 1:** FederatedAveraging. C is the global batch size; B is the local batch size; the K clients are indexed by k; E is the number of local epochs; and $\eta$ is the learning rate .

---

> **ServerAggregation:** // *Run on server*
>   initialize $w_0$
>   **for** each round $t = 1, 2, ...$ **do**
>     $m \leftarrow \max(C \cdot K, 1)$
>     $S_t \leftarrow$ (subset $m$ of clients )
>     **for** each client $k \in S_t$ **in parallel do**
>       $w_{t+1}^k \leftarrow$ **ClientUpdate** $(k, w_t)$
>     $w_{t+1} \leftarrow \sum_{k=1}^{K} \frac{n_k}{n} w_{t+1}^k$
> **ClientUpdate**$(k, w_t)$: // *Run on client k*
>   $B \leftarrow$ ( split dataset belong to $k$ into batches of size $B$)
>   **for** each local epoch $i$ from 1 to $E$ **do**
>     **for** each local epoch $i$ from 1 to $E$ **do**
>       **for** batch $b \in B$ **do**
>         $w \leftarrow w - \eta \nabla \ell(w; b)$
>   **return** $w$ to server

---

In one round (i.e., the t-th) of the FederatedAveraging, the server first broadcasts the newest model, $w_t$, to all the clients. Then each client (i.e., the k-th) make $w_t^k = w_t$ and then performs clientsupdates $E(\geq 1)$ times. Before the aggregation, each client's model should set a weight. It is mentioned above, the weight is the proportion of the client's dataset size $|n_k|$ to the total dataset size $|n|$. Finally, the server aggregates all local models to produce a new global model according to their different weights. Note that there exists the non-IID data and partial workers (clients) participation issues, the aggregation step would vary.

### 3.3. Blockchain

The blockchain was first proposed by Satoshi Nakamoto in 2009 [34]. As the public ledger of bitcoin, it plays an important role in data interaction. The blockchain is maintained by all user in a P2P-network. Transactions between users are packaged into blocks and stored on the blockchain. Due to the decentralized storage of the blockchain, the information on each transaction can be retrieved by anyone.

Ethereum is the blockchain environment used in our framework. In our experiment in Section 6, we constructe a private chain to realize the whole operation process of crowdsourcing, and show the experimental results in detail. Special note, Ethereum has the following two functions in our framework.

- **Distributed database.** Every node in the blockchain network keeps a copy of all transactions, and others can verify any transactions between two parties. This redundant storage strategy improves tamper-proofing of data and traceability by sacrificing storage space.
- **Righteous programmatic platform.** There are two types of accounts in Ethereum: the externally owned account controlled by private keys, and the contract accounts controlled by contract codes [35]. The subject of a contract account is a smart contract, which supports an external account to write the input content in the transaction data, and get the output result by calling a contract. A contract cannot be changed once published, so the creator must carefully examine the contents of the contract before publishing it.

*3.4. Secure Re-Encryption Scheme*

In traditional FL, the client usually sends the trained model to the server via Secure SHell (SSH). However, in the framework we design, all interactions are based on the blockchain, which means any participant could search each model in the ciphertext. At the same time, in order to implement the reward mechanism, REQ needs to judge each model on its own data and feedback judgment to CSP, then select available models for aggregation based on CSP scoring. To prevent unfair competition in scoring caused by malicious worker behavior, such as colluding with REQ, we design a new re-encryption algorithm.

ElGamal is an asymmetric encryption algorithm based on the Diffie–Herman key exchange. As an encryption algorithm with partial homomorphism, it has faster encryption and decryption speed than Paillier [36]. Based on the Elgamal encryption system, we design a secure re-encryption scheme suitable for the open data interaction mode of the blockchain. After some extensions, this algorithm can also be applied to other frameworks that also use the blockchain for data storage and interaction.

The secure re-encryption algorithm is consists of the following five subalgorithms:

1. $KeyGen(\mathbb{G}, p, g) \rightarrow \{pk_{csp}, sk_{csp}, pk_{req}, sk_{req}, PK\}$: In the key generation algorithm, $\mathbb{G}$ is a multiplicative cyclic group of a prime order $p$ with a generator $g$, such that discrete logarithm problem over the group $\mathbb{G}$ is hard. Then, the CSP and the REQ respectively choose a secret number as private key $sk_{csp}, sk_{csp} \in Z_p^*$ and compute $h_{csp} = g^{sk_{csp}}, h_{req} = g^{sk_{req}}$, then there exit two public key $pk_{csp} = (\mathbb{G}, p, g, h_{csp}), pk_{req} = (\mathbb{G}, p, g, h_{req})$. Moreover, we negotiate a special Diffie–Hellman public key $PK = (\mathbb{G}, p, g, h) = (\mathbb{G}, p, g, h_{csp} * h_{req}) = (\mathbb{G}, p, g, g^{sk_{csp}+sk_{req}})$.
2. $Enc(PK, m) \rightarrow \{c\}$: The encryption algorithm takes the public key $PK$ and a message $m \in \mathbb{G}$ as inputs, then it randomly chooses a number $r \in Z_p^*$ and output a ciphertext $c = Enc(PK, m) = (g^r, m \cdot h^r)$.
3. $ReEnc(sk_{csp}, c) \rightarrow \{\bar{c}\}$: CSP uses its private key $sk_{csp}$ for the re-encryption operation which is similar to common decryption process in Elgamal encryption system, and output a new ciphertext $\bar{c} = (g^r, m \cdot h^r / g^{r*sk_{csp}}) = (g^r, m \cdot h_{req}{}^r)$.
4. $Blinding(\bar{c}) \rightarrow \{|\bar{c}|\}$: The blinding algorithm takes a temporary rand number $r' \in Z_p^*$ as a rand factor and corresponding ciphertext $\bar{c}$ as inputs, and CSP outputs a new blind ciphertext $|\bar{c}| = (g^{r^{new}}, m \cdot h_{req}{}^{r^{new}}) = (g^r \cdot g^{r'}, m \cdot h_{req}{}^r \cdot h_{req}^{r'}) = (g^{r+r'}, m \cdot h_{req}{}^{r+r'})$.
5. $Dec(sk_{req}, |\bar{c}|) \rightarrow \{m\}$: REQ execute the decryption algorithm. It takes the $sk_{req}$ and blinding ciphertext $|\bar{c}|$ as inputs and outputs $m = m \cdot h_{req}{}^{r^{new}} / (g^{r^{new}})^{sk_{req}} = m \cdot h_{req}{}^{r^{new}} / (g^{sk_{req}})^{r^{new}}$.

It is worth mentioning that the blind ciphertext $|\bar{c}|$ and the ciphertext $\bar{c}$ have different ciphertext segments, that means anyone cannot recognize $|\bar{c}|$ from $\bar{c}$.

In the framework we designed, *Blinding* is essential for the model ciphertext interaction on the blockchain. It can ensure that any REQ who is biased against some workers would be absolutely fair in the aggregation process. The specific reasons will be explained in Section 5.

## 4. Preliminaries

*4.1. Overview*

Crowdsourcing is a sourcing method in which workers in the form of individuals or organizations obtain goods and services, including ideas and finances. In recent years, great progress has been made in various aspects of crowdsourcing, and many new applications using crowdsourcing have been proposed.

Since crowdsourcing occurs in the context of a complex online network trading, there are two main issues:

- **Quality issues.** Since crowdsourcing workers are usually Internet crowds, it is hard to ensure each worker will collaborate with the agreement. In some adversary environments, malicious

participants will violate the agreement and launch some attacks, such as malicious workers submitting a large number of low-quality solutions or stealing others' solutions and claiming to be theirs. These behaviors often lead to poor quality of crowdsourcing solutions.

- **Rewards distribution issues.** The ultimate purpose of workers is to get rewards by completing tasks. Some of the currently known distribution mechanisms will reward the worker who submitted the best solution, but this is unfair for other workers who have worked hard. The most reasonable profit distribution should be that any workers get the reward they deserve according to their own workload and the quality of the solution.

Generally, REQ focus on the quality of the solution, and workers hope for a more reasonable reward. In order to achieve a win-win consensus, we have redesigned a crowdsourcing framework that adopts blockchain network and FL algorithm. We call it CrowdSFL (secure-FL-based crowdsourcing).

The framework overview of CrowdSFL is illustrated in Figure 2.

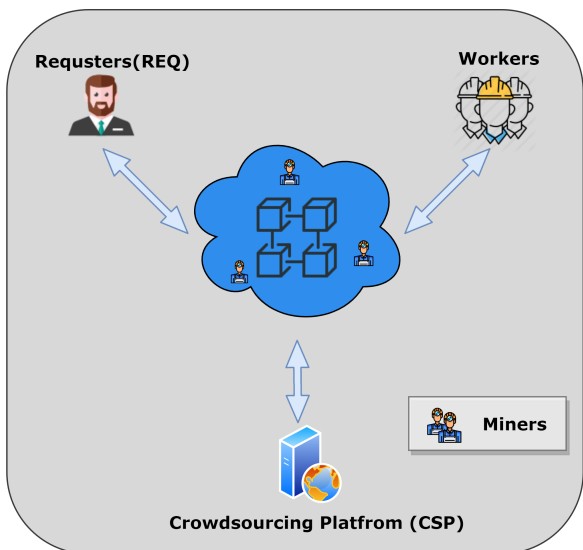

**Figure 2.** Overview of CrowdSFL. All participants interact via a P2P (peer-to-peer) network of blockchain.

Unlike traditional crowdsourcing, which selects the best solution from all the solutions, the solutions submitted by each worker in CrowdSFL will be selectively aggregated by CSP and finally affect the final crowdsourcing results to some extent. In other words, a practical solution comes from the effective efforts of most workers. For workers, the reward of each task will stimulate the initiative of workers, which also provides a new idea for the reward distribution mechanism of crowdsourcing. The focus of this paper is on the security and efficiency of crowdsourcing, and the rewards distribution will vary widely based on the task's environmental background, the reputation of the participants, the REQ's wealth and preferences, so no more discussion about the reward distribution mechanism would be provided.

### 4.2. Smart Contract

Smart contract, which refers to the Blockchain 2.0 space, is proposed by Nick Szabo in 1994 [37]. It empowers blockchain (e.g., Ethereum and Hyperledger) to perform complex operations by programming common process into code. Each Ethereum account can implement its logical functions by calling specific smart contracts. Due to the blockchain's peer-to-peer network and consensus mechanism, smart contracts can work without fraud, downtime, or tampering.

Different from traditional centralized crowdsourcing system, we implement our crowdsourcing system on Ethereum. Algorithm 2 illustrates the implementation of submitting data in the contract. The data interaction is implemented through the logic formulated by smart contracts to ensure that:

(1)    The data submitted by the participants conform to the correct format.
(2)    The data is stored publicly, and participants can retrieve the entire data from the blockchain.

---

**Algorithm 2:** Submitting Data.

---

    **(Function 1) SubmitMetadata**

    **(Function 2) WorkerSubmit:** // round $t$, worker $i$, worker's account number *account*

        **If** now $< T_{wait}$ **then**

            $C_{w_t^i}^{list} \leftarrow [account, t, c_{w^i}]$

    **(Function 3) CSPSubmit:** // after *Blinding*, ciphertext will be uniquely numbered as *ID*
    by CSP

        **If** $acc_t^{list} = NULL$ **then**

            $|C_{w_t^i}^-|^{list} \leftarrow [ID, t, |c_{w_t^i}^-|]$

        **else**

            $AvailableModelList \leftarrow [ID_a, ID_b, ...]$

    **(Function 4) REQSubmit:** // *acc* stands for model test accuracy results

        **If** $AvailableModelList = NULL$ **then**

            $acc_t^{list} \leftarrow [ID, acc]$

        **else**

            $NewModel \leftarrow w_{t+1}$

---

### 4.3. Model Data Interaction via Blockchain

In order to store encrypted model data on the blockchain, workers need to call smart contracts to create transactions. Through consensus mechanism algorithms, such as PoS, PoW, miners package multiple transactions into a new block. In Ethereum, theoretically, there is no direct or fixed upper limit on transaction size or block size, but this does not mean that there is no upper limit on the amount of data that a transaction can carry. GasLimit is a very important configurable parameter in Ethereum. It is the maximum amount of gas that users are willing to pay to execute a particular transaction successfully. Its being restricts the maximum data size that we can store in a block. In order to ensure the standardization of data interaction, we set the maximum value of the input data field in the contract not to exceed 64 KB. For the model size in general federated learning, this value is relatively redundant.

Based on the FederatedAveraging and our proposed re-encryption algorithm in Section 3, combined with smart contracts, we propose a model-data-interaction algorithm, the pseudo code is as follows.

Note that when performing new crowdsourcing, a new private chain will be created, and all data interaction will execute on the new private chain. Algorithm 3 guarantees CrowdSFL to protect the confidentiality and anonymity of the model, which can be said to be the core function of CrowdSFL. The design ideas of CrowdSFL are also based on this algorithm, and the specific framework architecture will be discussed in Section 5.

---

**Algorithm 3:** Model-Data-Interaction.

---

**Workers:**
**for** each worker $i \in k$ **in parallel do**
$\quad m_{w_t^i} \leftarrow$ **ClientUpdate**
$\quad c_{w_t^i} \leftarrow$ **Enc** $(PK, m_{w_t^i})$
$\quad$**Call the contract to upload** $c_{w^i}$ **to the chain**
**CSP:**
$\quad c_{w_t^i}^- \leftarrow$ **ReEnc**$(sk_{csp}, c_{w_t^i})$
$\quad |c_{w_t^i}^-| \leftarrow$ **Blinding**$(c_{w_t^i})$
$\quad$**Call the contract to upload** $|c_{w_t^i}^-|$ **to the chain**
**REQ:**
$\quad m_{w_t^i} \leftarrow$ **Dec**$(sk_{req}, |c_{w_t^i}^-|)$
$\quad$**If** $w_t^i$ is available, **then**
$\quad\quad w_{t+1} \leftarrow$ **ServerAggregation**
$\quad\quad$**Call the contract to upload** $w_{t+1}$ **to the chain**

---

## 5. Framework Architecture

In this section, we describe the architecture of our framework, including the system model that explains how crowdsourcing performs, and the threat model.

### 5.1. System Model

CrowdSFL consists of two parts, preliminaries before the crowdsourcing and the implementation of the crowd computing process. The working steps of the CrowdSFL are as follows:

**Step 1: REQ releases the tasks.**

Before the FL begins, the REQ needs to publish its basic requirements to the CSP, such as the functions of the solution it needs, what features the data should have, whether the model used for learning belongs to CNN, RNN or others, and all hyperparameters of the model structure, etc. Necessary information is crucial for workers to perform FL, and incorrect parameters or structure of model will affect or even destroy the efficiency of model aggregator. In a task that has a strict demand for accuracy, this means that there would be more training rounds. Since each data interaction is implemented on the blockchain, each worker participating in FL must store all interactive data on the blockchain locally.

In addition to the basic requirements, the REQ also needs to release the reward pool, which is the source of motivation for workers to participate in a computing task. A participation deadline is necessary. Before deadline arrives, workers can voluntarily choose whether to participate in a task released by a REQ on the CSP. Finally, the maximum waiting time $T_{wait}$ for a round of training needs to be set. In one iteration, the CSP will collect and process newly uploaded models on the blockchain after the maximum waiting time $T_{wait}$ ends. All workers must complete the training and upload new model within this specified time, otherwise, they will be judged to be sabotage in this round.

**Step 2: Workers voluntarily participate in tasks.**

Workers who want to get a reward, choose to participate in a task in a voluntary way. This is a common practice of crowdsourcing systems [15]. A malicious worker may wish to submit a wrong model to be rewarded which leads to model poisoning, so workers need to make a deposit before participation, which can effectively thwart many attacks such as DDoS, Sybil and "false-reporting" attacks. In FL, model poisoning can seriously affect the learning process [21].

The workers who have confirmed to participate in this task will start data interaction based on the blockchain. They firstly upload their metadata on the private chain in the form of transactions. Metadata contains the account number, the reputation of a worker, the size of the dataset $|n_k|$ owned

by the worker, and the hash value of the dataset. The account number is a unique identifier of the worker. Reputation is a value representing workers' participation. The higher the reputation, the more tasks this worker has successfully completed, in a sense, he is a veteran.

The deposit submitted by the worker is a function based on the reputation and $|n_k|$. The higher the reputation, the fewer deposits he needs to submit. The larger $|n_k|$, the more deposits he needs to submit, because in Algorithm 1, the calculation result of the model aggregator will be biased to the worker with a large amount of data. When the size of a dataset owned by a worker almost occupies the entire dataset size, likely 99%, the result of model aggregation will be approximately equal to the model submitted by this client, so he needs to submit more deposits to ensure that he is a positive worker. Similarly, the worker can finally get reward as a function of $|n_k|$. A larger $|n_k|$ means that the worker has more resources and has exerted more power in the construction of the classifier. From the perspective of "data is wealth", after the training process, a worker with more data deserves more rewards.

The configuration of deposits and rewards is variant between different scenarios, depending on the nature of a crowdsourcing task, which is not the focus of this paper.

### Step 3: Contract deployment.

Through the code constructed in the contract deployed, workers, REQ, and CSP can implement uploading and querying. In the concept of Ethereum, the storage operation of the blockchain is quite expensive, each account node would save a copy of the data locally, so the participants will not upload oversized data, for instance, one dataset of HD pictures. Moreover, Ethereum supports that query the information of the blockchain without requiring any transaction fee since the information has been synchronized locally due to the blockchain transaction mechanism. In the following operations, all data interactions are essentially repeated calls to this contract.

### Step 4: Model initialization.

The REQ generates an initialization model $w_0$ randomly and uploads $w_0$ by calling the contract.

### Step 5: Querying, Training and performing Enc.

Workers query the model from blockchain and training with their own data locally. During the $t$-th round, each worker $k$ that normally participates in training can calculate $w_t^k$ and then generate a ciphertext $c_{w_t^k}$ by the encryption key $PK$. Finally, $c_{w_t^k}$ is uploaded to blockchain before the end of setting $T_{wait}$.

### Step 6: CSP performs ReEnc and Blinding.

When $T_{wait}$ arrives, CSP queries new transactions information on the blockchain and obtains all $c_{w_t^k}$ from it. It first performs *ReEnc* and *Blinding* to get $|c_{w_t^k}^-|$. To achieve a fair scoring mechanism, $c_{w_t^k}$ and $|c_{w_t^k}^-|$ are backed up locally on the CSP. Finally, each $|c_{w_t^k}^-|$ and their unique ID are uploaded to the blockchain.

### Step 7: REQ performs Dec, judges model accuracy.

REQ query the blinding ciphertexts and performs *Dec*. Since the blinding ciphertexts $|c_{w_t^k}^-|$ have different ciphertext segments to $c_{w_t^k}$, a malicious REQ who is biased against some workers cannot figure out a blinding ciphertext belongs to which worker, the only thing REQ can do is performing the operation in the protocol.

When REQ gets all plaintexts of model $w_t^k$, he needs to judge the accuracy on his own dataset and upload the accuracy corresponding to each blinding ciphertext to the blockchain.

### Step 8: CSP scores and selects available models.

Now the CSP has the accuracy of each worker's model, as well as the metadata of each of them, and then scores according to the established rules. Finally, he will publish scores and select the ID corresponding to each available model in ciphertexts to upload to the blockchain.

### Step 9: REQ performs aggregation.

REQ selects the corresponding model in plaintext to participate in the aggregation. Then, the result $w_{t+1}$ would be uploaded to the blockchain.

**Step 10: Return to step 5 and into the next round.**

The execution of one round ends and the next round of training will be performed until the stop condition is reached. Figure 3 illustrates all the working steps of the CrowdSFL.

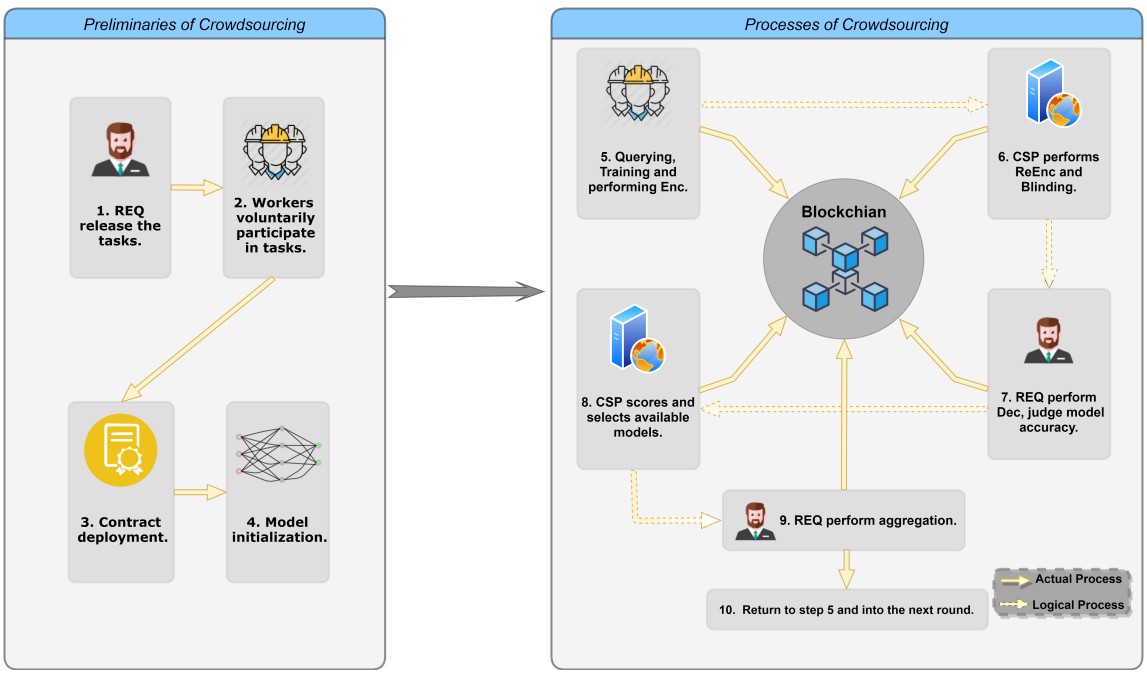

**Figure 3.** Overview of CrowdSFL.

*5.2. Evaluation Mechanism*

In Step 7 and Step 8, REQ and CSP will work together to obtain a list of models that will participate in the aggregation. This corresponds to solution evaluation, and the difference is that the evaluation result is not to choose the best solution but to score one by one. For REQ, it is hoped that only the models with high accuracy values will participate in the aggregation, in order to finally get a solution that is sufficiently robust to all data spaces. For workers, they hope that they can get higher scores as much as possible to participate in aggregations and finally get more rewards. Therefore, REQ and workers in the evaluation process can be considered from the perspective of game theory.

Wu et al. [38] proposed an evaluation framework for software crowdsourcing. They modeled and quantified software quality, costs, diversity of solutions, and other attributes in software crowdsourcing. They proposed participation-output analysis, submissions-award analysis, and some other analysis methods to evaluate software crowdsourcing. Affected by it, we propose a scoring evaluation mechanism based on min-max analysis.

When CrowdSFL is working, CSP has the size of each worker's dataset, reputation value, and the accuracy value of the model submitted in each round of training. In traditional machine learning, a dataset with high quality and large amounts will produce a better model. In a round, if a worker with a large amount of data submits a model with low accuracy, it can be inferred that the worker is "negative". In our scoring evaluation mechanism, we set the CSP to rank the dataset size of each worker and the model's accuracy value in each round, which are $X$ and $Y$, respectively. The difference value between the two is the result of scoring.

$$Score = X - Y \tag{1}$$

The larger $X$ means that the worker has more data, so REQ will hope that the model he submitted can have higher accuracy. The larger the $Y$, the better the model submitted by the worker, and the worker will hope that he will get a higher score by submitting the solution. $X - Y$ represents the game result of REQ and worker in a round. When the score is smaller, the REQ is more dominant, and when the score is larger, the worker is more dominant. Considering that in the crowdsourcing system, the reputation value can represent the potential positivity of the worker, the revised scoring evaluation function is:

$$Score = X - Y + W \times R \tag{2}$$

$W$ represents the weight of reputation value in the scoring evaluation mechanism, and it can set as needed in different crowdsourcing.

The score can not only help CSP to select the model to participate in aggregation, but also provide a reference for the reward distribution mechanism of crowdsourcing. The main goal of this paper is the security and efficiency of crowdsourcing. Reward distribution needs to consider more realistic factors, and there are few general reward distribution mechanisms in previous related works, so no more introduction about it is made in this paper.

### 5.3. Threat Model

We define privacy as all kinds of information submitted by participants to the blockchain, to ensure that:

1.  All ciphertexts cannot be decrypted.
2.  Participants cannot find the rules of the information on the blockchain. For example, REQ can find the corresponding gradient plaintext through the blinded gradient ciphertext.

Potentially malicious participants may take different behaviors to maximize their own profits [15]. We first define the threat model, which describes the potential threats and malicious behavior as follows:

*   **Malicious REQ.** REQ should release the reward pool he provided before performing the task. Considering that the purpose of malicious REQ is to obtain a useful solution and reduce his asset consumption, he may use some features of crowdsourcing to achieve it. For example, malicious REQ will recruit some fake workers to participate in crowdsourcing, and misrepresent the solutions submitted by these workers as a high-quality solution. After crowdsourcing, these workers recruited by malicious REQ can theoretically receive more rewards, so that malicious REQ can recover some assets from the reward pool.
*   **Malicious Workers.** Malicious workers attempt to obtain rewards without paying sufficient effort, which is free-rid-ing attack. Workers who scored low during the evaluation would want to change these in other ways, such as denying or even fork the existing blockchain. In addition, malicious workers can directly grab other workers' solutions and claim to be their own, so as to achieve reaping without sowing.
*   **Malicious Miners.** Malicious miners attempt to disrupt the normal execution of programs on the blockchain by forking chains or collaborating with malicious participants, thereby achieving their attack goals.

We formalize the entire process of crowdsourcing as completely secure and fair, and give the following definition:

*Definition. (Completely Fair and Collusion Resistance)* Each worker will get the corresponding due reward by the amount of work he paid out, and REQ will get a relatively objective and available solution if all participants are semi-honest. That is, each participant follows the implementation process of the agreement, but the participants will try to infer the information of others based on the public information on the blockchain. The Elgamal encryption system guarantees the confidentiality of the information. Assume that malicious REQ uses brute-force-guessing to find information rules. When the number of gradient ciphertexts on the blockchain is $x$, the probability of req to accurately

obtain the plaintext corresponding to all ciphertexts is $1/x!$. Every round, REQ needs to re-guess. The computational complexity of the brute-force-guessing would be $O(r * X!)$.

Malicious participants always violate the rules to achieve their own purposes, but after that, CSP could easily find the malicious participants through auditing, since all the interaction data is recorded on the blockchain. For a malicious participant, CSP will confiscate his deposit and reduce his reputation. Given that CSP is generally a credible institution, he is generally credible and honest.

## 6. Performance Evaluation and Analysis

In this section, we design an experiment to apply CrowdSFL to a crowdsourcing. In the experiment, we assume that all participants are semi-honest, and show the superiority of CrowdSFL by comparing with SecureSVM in [16]. It also adopt the blockchain as a platform for data interaction. Unlike CrowdSFL, SecureSVM [16] encrypts the scattered datasets of multiple IOT devices with Paillier, and uses the homomorphic characteristics of Paillier to apply the designed ScureSVM algorithm to the data in ciphertext. However, in real-world scenario, it is unwise to store a large amount of data on the blockchain. The size of a complex dataset often exceeds the range that the blockchain can bear, and [16] is to store the ciphertext generated after data encryption into the blockchain. The size of the ciphertext can be much larger than the plaintext, which adds a greater burden to data storage.

### 6.1. Experiment Design

In our experiments, we use CrowdSFL to implement a crowdsourcing: a prediction task is issued by REQ, and all participants cooperate to complete this task securely. The experiment was run on a PC equipped with a 6-core I7-8750H processor at 2.20 GHz and 24 GB RAM. Since each worker independently performs gradient calculation and encryption on the data it owns, and the blockchain is a private chain maintained by all participants, we use Truffle (Truffle Blockchain Group, Inc. Open source on GitHub, v2.1.1) to simulate the Ethereum private chain. SYFT (Open source on GitHub, v0.2.1) framework virtualizes the workers and REQ in the crowdsourcing. The host acts as CSP. We have implemented the re-encryption, the FL algorithm in Python 3.6.

The goal of the experiment we designed is the same as in [16], to achieve a classification prediction. For better comparison experiments, we also use two real-world dataset which named Breast Cancer Wisconsin Data Set (BCWD) and Heart Disease Data Set (HDD). BCWD contains 699 instances, each instance has 9 numerical attributes. The features of BCWD are computed from a digitized image of a fine needle aspirate of a breast mass and describe characteristics of the cell nuclei present in the image. Each instance is labeled as benign or malignant. HDD contains 303 instances, each instance has 13 discrete attributes. We randomly shuffle the two datasets, and similarly select 80% as the training set and the remaining 20% for testing.

One difference between CrowdSFL and SecureSVM [16] is that SecureSVM [16] chooses Paillier as the encryption scheme, and CrowdSFL uses Elgamal as the encryption scheme. ElGamal's security comes from the Diffie–Hellman problem and the underlying discrete logarithmic assumption. Paillier's security stems from the computational difficulty of the $n$-th residual degree, which leads to the Decisional-Composity-Residuosity-Assumption [36].

Considering that there are two simple binary classification prediction, We divide the dataset into $k$ and set the number of workers $k$ to 3, the number of training round $t$ to 5. According to the characteristics of the two datasets, we establish appropriate network structures and parameters respectively.

In order to prove that CrowdSFL protects the privacy and security without reducing accuracy, we additionally adopt a three-layers BP neural network with the same structure and parameters on a single machine to implement this classification prediction. Table 1 records the test results.

**Table 1.** Test accuracy.

| Dataset | CrowdSFL | BP-NN | SecureSVM [16] |
|---------|----------|-------|----------------|
| BCWD | 91.42% | 94.29% | 90.35% |
| HDD | 93.94% | 95.65% | 93.89% |

From the table we can see that CrowdSFL is better than [16] in prediction performance.

### 6.2. Time Overhead

The time overhead for CrowdSFL to perform a crowdsourcing is proportional to the number of rounds $t$. In general, when each workers has submitted a solution or the wait time $T_{wait}$ ends, the CSP will execute step 6. All the factors that affect the time overhead in a crowdsourcing include: (1) FL executing, (2) the re-encryption algorithm executing, (3) transactions packaging by miner, and (4) scoring and selecting by CSP. The time overhead of (4) will vary according to the complexity of crowdsourcing. Since it has been explained earlier that the number of workers in this experiment is small and all are semi-honest, the time overhead of (4) can be ignored. Shen et al. [16] gives the time overhead for calculations in homomorphic encrypted ciphertext. We compared the time overhead of CrowdSFL on BCWD with [16], and found that the time overhead of CrowdSFL is much smaller than [16]. The results are shown in Figure 4.

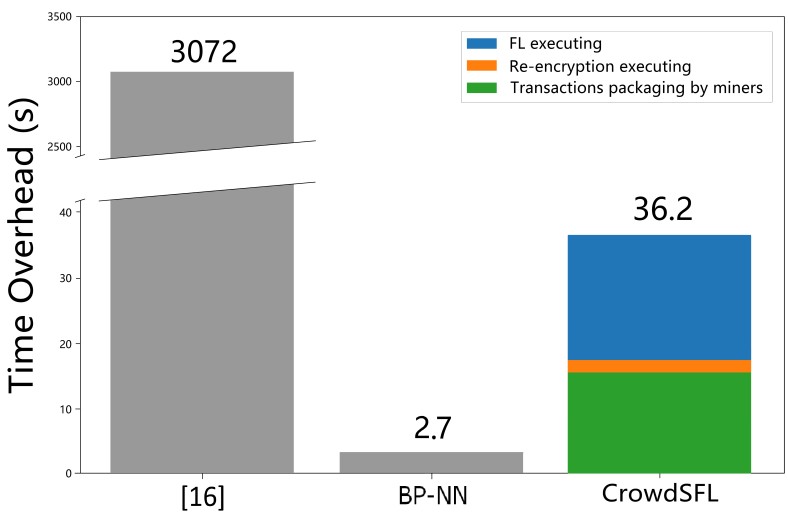

**Figure 4.** The time overhead of CrowdSFL, BP-NN, and [16].

Since CrowdSFL does not involve the operation of homomorphic ciphertext, it takes much less time to complete the same task in [16]. In this context, we carry out experiments repeatedly, and it took 36.2 s to achieve a crowdsourcing on average, including: (1) 18.20 s, (2) 2.18 s, and (3) 15.82 s. It should be noted that since the private chain in this experiment was created by Truffle, each new transaction will be quickly reacted and packaged by the miner. In a real scenario, the time overhead of (3) is related to the network bandwidth of each user. When a participant's network is congested, it may bring bad effects.

Both [16] and CrowdSFL work on the blockchain P2P network. The distributed system always shows poor performance due to additional network communication, computing interaction, and other real-world conditions. To objectively show the time overhead of CrowdSFL, we complete the same task with BP neural network on a single machine, and the total training time of 5 rounds takes 2.7 s. From the experimental results, we can see the CrowdSFL would be an order of magnitude higher in

time overhead than single machine computing. Still, this is completely reasonable and compared to [16], which also works on the blockchain, the time overhead required by CrowdSFL is more acceptable.

## 6.3. Communication Overhead

Blockchain is a decentralized distributed system. Its integrity and security guarantees benefit from its consensus mechanism [34], but this is obtained at the cost of each node on the network saving the entire blockchain information. It is quite unwise to store a large amount of data on the blockchain, because each node will download the newly added data to the local.

If there is a dataset with a large number of instances (far more than 699 instances in BCWD) that need to be encrypted and stored in the blockchain. This should not only consider the feasibility, but also the practical significance of the method. In CrowdSFL, the size of the model is generally much smaller than the size of the dataset, which is mainly determined by the neural network structure. In this experiment, we take BCWD as an example to record the size of the plaintext and ciphertext in each upload in a crowdsourcing, as well as the gas cost of completing a prediction task in Ethereum. Then we compare it with the theoretical data size of [16], and the detailed results are shown in Table 2. Note that the size of ciphertext and the gas cost of [16] is a theoretical minimum in [36].

**Table 2.** Comparison of communication overhead between CrowdSFL and SecureSVM [16].

| Scheme | Size of Plaintext | Size of Ciphertext | Gas Cost |
|---|---|---|---|
| CrowdSFL | 331 B | 690 B | 455,016 gas |
| SecureSVM [16] | 14.9 KB | 29.8 KB * | 2,013,336 gas * |

As can be seen from Table 2, uploading the model to the blockchain is relatively wise compared to uploading the total datasets. Even though there are multiple data interactions in FL, from the perspective of gas cost, CrowdSFL is an order of magnitude stronger than the [16]. In Ethereum, the gas cost of storing is quite expensive, the unit price of each ETH is as high as $170 in early April 2020, the method adopted in [16] is almost unaffordable to apply in real situations.

## 6.4. Analysis

From the above experiments, we can see that the CrowdSFL is suitable for completing computing tasks with large numbers of data instances. The emergence of FL breaks the limitation of data centralization in traditional distributed systems. Nearly all the processes of CrowdSFL completing crowdsourcing have been transformed into the interaction of ciphertext. In contrast, [16] needs to collect data from each device before performing calculations. This will pose a more serious threat to privacy when facing an untrusted analysis center.

In one crowdsourcing, workers, REQ, CSP, and miners need to work together. All operations are divided into synchronous operations and asynchronous operations. Each worker will perform the gradient update and encryption process independently, so step 5 is an asynchronous operation. CSP immediately performs re-encryption and blinding operations on each ciphertext model. The blinded model is given a unique ID and re-uploaded to the blockchain, so step 6 is also asynchronous. REQ performs decryption and judgment process, so step 7 is also an asynchronous operation. The premise of scoring in step 8 is all the judgment results in step 7. At the same time, the aggregation in step 8 requires step 7 to provide a list of available models. These two parts are the synchronous operation.

It is feasible to use the blockchain as an interaction platform for the ciphertext model. The size of the data that needs to be stored and communicated in the blockchain for crowdsourcing is mainly related to the number of FL round $t$, the number of workers $k$, and the size $S$ of the model.

In a round, the ciphertext of the trained model will be uploaded by each worker firstly. The CSP processes all the ciphertexts and uploads them again. Finally, the new aggregation model will be uploaded to the blockchain in the form of plaintext, and the next round will be started. Generally

speaking, the data size of the ciphertext is more significant than that of the plaintext. The growth of the size is closely related to the encryption system adopted. Chen N. [36] gives the comparison of the ciphertext size changes between Elgamal encryption and Paillier encryption, and it states that both encryptions would multiply the size. Then the total data size that CrowdSFL will upload to the blockchain in crowdsourcing can be represented as

$$S_{CrowdSFL} = (k \times 2S_{model} + k \times 2S_{model} + S_{model}) \times t = (4k + 1) \times t \times S_{model} \tag{3}$$

The size of interactive data in CrowdSFL $S \propto (4k + 1) * t$. In order to save the total overhead and every individual participant's gas cost, the appropriate training round $t$ is very important.

In [16], the data of multiple IOT devices is aggregated and encrypted by Paillier and uploaded to the blockchain. The total data size can be represented as

$$S_{[16]} = 2 \times S_{dataset} \tag{4}$$

Generally, the size of the dataset is much larger than the size of the model, and we can clearly find some related data from the experiment results in Table 2. Through some model compressions schemes such as [39,40], we can further reduce the model size. The BCWD and HDD datasets in the experiment are in plain-text format. For more complex image datasets, such as MNIST handwritten digits datasets, the file size of the dataset is several orders of magnitude higher than the size of the model, and the feasibility of [16] will be significantly reduced. This negative effect will become more obvious as the complexity of the dataset increases.

## 7. Conclusions and Future Work

This paper innovatively proposes a crowdsourcing framework CrowdSFL addressing the problem of security issues in crowd computing. Our work combines blockchain, federated learning, and re-encryption technology to build up a decentralized and secure crowdsourcing framework. The blockchain serves as a network environment for data interaction. All blockchain nodes, who are also participants in crowdsourcing, will be able to retain a full data backup to improve the auditability of crowdsourcing. Considering that the blockchain is not suitable for storing massive data, we adopt federated learning, and each worker can upload the model or gradient data to REQ for aggregation. In addition, in order to conquer the malicious behaviors in crowdsourcing, we designed a new re-encryption algorithm to ensure the fairness of evaluation and scoring.

In the experiment in Section 6, we analyze the performance of CrowdSFL from various perspectives. Compared with similar representative work [16], we find that our work is better than the result of [16] in terms of accuracy, but compared to the BP-NN under the single machine, our result value is still slightly worse. We can probably speculate that this is due to the loss of information caused by the low precision of model values when federated learning performs gradient aggregation. However, this performance degradation is acceptable for the overall framework. In terms of time overhead, CrowdSFL has a considerable advantage. The time required to complete a prediction task is 1/85 of [16]. In terms of communication overhead, the gas cost required by the uploading model is much smaller than the way of uploading the entire data set to the blockchain in [16], which would make our framework more practical.

At present, our framework is not perfect enough. In the future, we must think about the establishment of a reward distribution mechanism. The reward distribution mechanism and reputation value adjustment would change in different scenarios, so we will establish the basic rules of both in future work and develop them through smart contracts. We have shown through experiments that CrowdSFL has an excellent performance in achieving prediction tasks on BCWD and HDD datasets, and is far more reliable than some similar work. However, as prices rise, the cost of implementing model interaction on Ethereum will also increase, so CrowdSFL also needs to be further optimized. In some more complex real-world scenarios, such as CNN for image classification or RNN for semantic

recognition, a complex model means uploading more data to the blockchain, so we need to optimize the practicality of CrowdSFL. For example, we would use the model compression technology in [39] to further condense the size of the model.

The development of blockchain technology and FL algorithms has brought new energy to traditional computing crowdsourcing. Our work is one of the first approaches that combine federated learning with crowd computing by making use of blockchain technologies. Today, when crowd computing has not yet reached its climax, our work has contributed cases and ideas for the future development of technology and applications in crowdsourcing.

**Author Contributions:** Z.L. contributed to writing—original draft preparation, methodology, software, and validation of the proposed scheme; J.H. contributed to conceptualization; J.L. contributed to writing—review and editing and funding acquisition; H.W. contributed to supervision; and M.X. contributed to project administration. All authors have read and agreed to the published version of the manuscript.

**Funding:** This research was funded by the National Natural Science Foundation of China under Grant No. 61801489.

**Conflicts of Interest:** The authors declare no conflict of interest. The funders had no role in the design of the study; in the collection, analyses, or interpretation of data; in the writing of the manuscript, or in the decision to publish the results.

## Abbreviations

The following abbreviations are used in this manuscript:

| | |
|---|---|
| P2P network | Peer-to-Peer network |
| BP-NN | Back Propagation Neural Network |
| CSP | Crowdsourcing platform |
| REQ | Requester |

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
