# Peer review of "CrowdSFL: A Secure Crowd Computing Framework Based on Blockchain and Federated Learning"

_electronics, doi:10.3390/electronics9050773_

Round 1
Reviewer 1 Report
This solution proposal paper is motivated by the problem to solve the security and privacy issues encountered in traditional crowd computing while ensuring that crowdsourcing is efficient enough.
The authors propose a blockchain-based decentralized framework (CrowdSFL) that uses re-encryption and federated machine learning methods. It might be one of the first approaches that combine federated machine learning with crowd computing by making use of blockchain technologies.
The validation of the approach is compared with a study dealing with privacy-preserving support vector machine training over blockchain-based encrypted IoT data on a simulated private Ethereum blockchain. According to that, the approach is magnitudes more performant. However, the question arises whether there are no more and comparable approaches that should be compared.
For instance, the authors did not compare their approach with approaches being not relying on blockchains. Because of the generally poor performance of blockchains, it might be not so outstanding compared with other volunteer computing projects like Folding@home and more.
It would be interesting to see the performance comparison between volunteer computing systems and the proposed one. So, the authors should consider the performance of volunteer computing projects and relate their solution proposal to this kind of systems as well to provide the reader with a broader picture of the expectable performance. Furthermore, it would be a good academic practice to report on the limitations of the presented approach.
Except for these optimizable points, the paper seems to present an interesting topic that is worth to be investigated more intensively by the research community.
Author Response
Thank you for your review. I have summarized your comments into 2 key points and completed the responses one by one. Thank you for pointing out that experiments related to volunteer coputing could be supplemented. I stated my thoughts and reasonably supplemented the experiment. Detailed responses are all recorded in the attachment. Sincerely, please see the attachment.

Reviewer 2 Report
1. The idea is interesting.
2. the structure of the paper as well as the names of the sections need improvement. For example, section 2 is called Related work and section 3 background. The latter could change because it confuses.
3. The text should be proof-read by a native speaker and correct all accidental slips or errors.
eg in the introduction …that require the a lot of calculations,…
4. The results of section 6, are partially analysed but not discussed properly. The analytical discussion of the results should be added in section 6 and the conclusions drawn should be discussed further in section 7.
5. Section 7 does not present a good discussion of the results in combination to the results of the performance evaluation. Furthermore, section 7 should present the contribution of this work.
Author Response
Thank you for your review. I have summarized your comments into 4 key points and completed the responses one by one. If you still have doubts about it, please be sure to tell me, I will continue to follow up the modification until your approval. Sincerely, please see the attachment.

Round 2
Reviewer 2 Report
The paper is improved. however, I still think that needs a native speaker reading because some sentences seem a bit weird.
The discussion on the conclusions section should be more extensive focusing on the analysis of the results.
Author Response
Your comments let us reflect deeply on the optimizations we made in the first revised version. We believe that there are still many areas for improvement in language organization. So we carried out more in-depth retouching and highlighted all the changes made in the second revision. In view of the many modifications, we have prepared a form for you to check more easily.
Please see the attachment. Sincerely.
